# Review of Small Gas Turbine Engines and Their Adaptation for Automotive Waste Heat Recovery Systems

**Dariusz Kozak *** and Paweł Mazuro

Department of Aircraft Engines, Warsaw University of Technology, 00-665 Warsaw, Poland; pmazuro@itc.pw.edu.pl
* Correspondence: dariusz.kozak.dokt@pw.edu.pl; Tel.: +48-509-679-182

**Abstract:** Current commercial and heavy-duty powertrains are geared towards emissions reduction. Energy recovery from exhaust gases has great potential, considering the mechanical work to be transferred back to the engine. For this purpose, an additional turbine can be implemented behind a turbocharger; this solution is called turbocompounding (TC). This paper considers the adaptation of turbine wheels and gearboxes of small turboshaft and turbojet engines into a two-stage TC system for a six-cylinder opposed-piston engine that is currently under development. The initial conditions are presented in the first section, while a comparison between small turboshaft and turbojet engines and their components for TC is presented in the second section. Based on the comparative study, a total number of 7 turbojet and 8 turboshaft engines were considered for the TC unit.

**Keywords:** auxiliary power unit; waste heat recovery; turbine engines; turbine wheel; opposed-piston engine; internal combustion engine; review

## 1. Introduction

High-temperature exhaust gases from internal combustion engines (ICE) offer the potential for heat recovery systems. Mechanical turbocompounding (mech-TC) or electric turbocompounding (el-TC) systems have been used in heavy-duty diesel (HDD) engines. Recently, a few interesting studies were conducted on TC.

Jääskeläinen and Majewski [1] carried out a review of TC systems for HDD engines; engine manufacturers such as Scania, Volvo, Daimler MB, Detroit, and Cummins have used serial TC systems in their engines. Axial turbines were used for the Volvo D13TC, MD13TC, and D12 500TC models, the Daimler OM 478 model, and the Detroit DD15 model. The implementation of TC resulted in a 3% efficiency improvement. Radial turbines were used in the Scania DT12, DTC11, and DTC12 models and the Cummins NTC-400 model; they resulted in a 4.5% efficiency improvement.

The history of Scania engines has been presented in the literature [2]. The author mentions positive aspects of TC in Scania DT12 engines and that such an engine meets the Euro 5 emission regulations.

In the literature [3], the available energy of the waste heat from a 6-cylinder HDD engine was analyzed. This engine was equipped with a high- and low-pressure loop exhaust gas recirculation (EGR) system. The authors compared three types of waste heat recovery systems, while 1-stage and 2-stage turbocharging systems were compared against a turbocharging system using the Rankine combined cycle (RCC). At full-load, the 1-stage turbocharging system provided 40.0 ÷ 74.8 kW of available energy from the waste heat, whereas the 2-stage turbocharging system provided 43.0 ÷ 76.3 kW of available energy. At low-load conditions, the available energy was small compared to high-load conditions. It was also found that turbocharging with the RCC improved the brake-specific fuel consumption (BSFC) by about 3%.

Ismail et al. [4] conducted a 0-D simulation on a 2 L diesel engine with a series TC unit. By locating the power turbine behind the turbocharger, the authors showed that the BSFC could be reduced; this was caused by the increase in the engine's back-pressure. They also showed the potential of TC combined with an EGR valve; this combination resulted in a power gain of about 14 kW considering isentropic expansion.

Furthermore, in their next work, Ismail et al. [5] investigated the effect of parallel TC on a 2 L diesel engine. It was shown that an engine model with parallel TC provided lower energy recovery potential than series TC. For low-load conditions, parallel TC had a negative effect on the global power gain, as most of the exhaust gasses passed through the turbocharger instead of the recovery turbine. At high load, a minor increase was observed for parallel TC; however, the authors showed the potential of parallel TC, as it produces lower back-pressure compared to series TC. In addition, in order to improve parallel TC, the authors suggested implementing a variable turbine geometry (VTG) in the turbocharger. With proper optimization of the VTG and reconfiguration of the turbocharger to run at higher efficiency, it is possible for the engine to benefit from parallel TC.

Experimental tests of parallel TC were also performed on an IVECO 3 L engine with two exhaust configurations [6]—the first was with the original VTG turbocharger and the second was with a new fixed geometry turbocharger. The first configuration showed good results for the engine at low loads, however at high loads the advantages were minor compared to the fixed geometry turbine. However, both systems displayed potential for power recovery of about 2% ÷ 2.5%.

Bin et al. [7] designed and tested a low-pressure turbine for TC applications in a 1.0 L gasoline engine. They conducted 1D simulations on the 1.0 L gasoline engine using three configurations of the low-pressure turbine—post-catalyst, pre-catalyst, and waste-gated configurations (in which the low-pressure turbine was located in a parallel configuration to the turbocharger). In low-speed, high-load conditions, the post-catalyst turbine position proved to be the best, with a reduction in the BSFC of 2.38% and an increase in the brake mean effective pressure (BMEP) of 2.4%. The pre-catalyst turbine position resulted in less of an improvement in both the BSFC and BMEP. This was caused by an increase in the back-pressure, thus causing higher pumping losses. The waste-gated, pre-catalyst turbine position had a negative effect on the engine's performance at low speed. This was mainly due to the small bypass ratio of the main turbine used to feed the low-pressure turbine due to its parallel configuration.

A comparison of an ICE diesel engine with TC against a turboshaft engine used as the propulsion system for a light rotorcraft was carried out in the literature [8]. The authors concluded that the ICE diesel engine offered a promising reduction in both fuel consumption and $CO_2$ emissions, especially with medium and high loads and for long-range missions. In addition, implementing the TC system further increased the reduction in fuel consumption. The reduction in fuel consumption of 25% ÷ 50% compared to the turboshaft engine depended on the required power. The higher mass of the ICE was compensated by the lower fuel consumption compared to the turboshaft engine.

He and Xie [9] investigated three modes of TC, namely electric TC, series TC, and parallel TC, on a 6-cylinder HDD engine. They performed simulations at a steady-state for different engine loads and speeds. In addition, three different driving cycles were investigated with these modes. The results showed that electric TC provided the best power increase at high-load and at a high-speed engine operating point. The best reduction in fuel consumption was achieved by electric TC at high-load and a high-speed operating point. The smallest power increase and reduction in BSFC were obtained from parallel TC. All the TC systems worked well for high-load and with a high-speed engine operating point. In different driving cycles, electric TC provided the best BSFC reduction of more than 7%.

Bin et al. [10] developed and tested a low-pressure turbine for electric TC applications. A TC turbine was designed for a very low-pressure ratio in order to reduce the impact of the back-pressure on the engine's performance. Tests with electric TC were performed on a 1.0 L EcoBoost gasoline engine in the series configuration. At 2500 rpm, a maximum reduction in the BSFC of 2.6% was achieved.

Salehi et al. [11] studied the influence of electric TC on a 13 L HDD engine with an asymmetric twin-scroll turbocharger. The first approach of simply implementing electric TC downstream of the exhaust system resulted in a reduction in the BSFC and produced smoke emissions. This was a result of the increase in the back-pressure and higher pumping losses. After modification of the turbocharger's volute and exhaust pipes to achieve a symmetrical volume, the BSFC was reduced by 1% at medium speed and load. Additionally, the transient operation of electric TC with a bypassed valve provided a 1.6% reduction in the BSFC.

Although Yamaguchi et al. [3] revealed a greater potential for turbocharging when combined with the organic Rankine cycle (ORC), such a solution has not been considered in this paper due to its complexity and its greater increase in mass. The articles have shown the potential of series TC as a waste heat recovery system. Parallel TC offers small benefits for power recovery and BSFC reduction. Additionally, in order to control the back-pressure in the exhaust system, and thus emissions, a VTG turbine should be introduced into the exhaust system.

A 2-stage mechanical TC system will be implemented in the 6-cylinder opposed-piston engine that is under development at Warsaw University of Technology. The recovered waste heat will be returned to the crankshaft by means of the gearbox.

## 2. Review of Potential TC Solutions

After World War Two, increasing demand for fast travel forced engine manufacturers to increase the brake power of their engines. Hot exhaust gases from an ICE promised a great source of energy for increasing an engine's power. Starting from the 1940's, the first engines were equipped with an exhaust turbine coupled with the engine's crankshaft via the gearbox. Despite the fact that the engine's brake power was increased, a great deal of energy was wasted on driving the exhaust turbine. The use of hot exhaust gases as the energy source for the turbine improved the engine's overall efficiency, and thus the engine's brake power. This application was used in both aircraft and automotive engines. Further demand for increased aircraft speed and a low power-to-weight ratio for piston engines made turbine engines more promising as a propulsion system. Turbine engines were first used in aircraft, however there was consternation about the use of such engines in the automotive industry. Unfortunately, the difficulty of controlling such engines and the high fuel consumption made them inappropriate for automotive applications. However, rotary equipment is still widely used in piston engines for waste heat recovery. In this article, small turbojet and turboshaft engines that are available on the market, along with their rotary equipment, were considered as a 2-stage TC system for the 6-cylinder OP engine, PAMAR V, which is under development. This is one of the options, aside from designing a new radial turbine. The reason for this is that adapting turbine engines that are already on the market is still cheaper than designing a new impeller. Designing a new turbine is also an option, however it has not been discussed in this paper. Due to the highly isolated combustion chambers in the previous engines, namely PAMAR III and PAMAR IV, high exhaust temperature is available. Thus, extremely temperature-resistant turbine components need to be used, especially in the first stage of TC. A review and comparison of small turbojet and turboshaft engines was carried out, considering the turbine inlet temperature (TIT) $T_{in}$, the maximal mass airflow $\dot{m}_{air}$, the maximal rotational velocity $n_{max}$ or revolutions per minute (RPM), the bearing system, and the gearbox's gear ratio.

### 2.1. Review of Small Turbojet Engines

As stated in the literature [12], from the late 1960's there was an increased interest in smaller turbojet engines. Engine manufacturers such as Teledyne Turbine Engines, Williams International, Turbomeca, Noel Penni, and Microturbo were leaders in the market. However, the main disadvantage of turbojet engines is the lack of an output shaft for transferring the mechanical power to other devices.

The Teledyne J402 was a single-shaft turbojet engine, with a maximal TIT of 1116 K, a maximal RPM of 41,200 r/min, and a maximal thrust of 2.9 kN [13].

Bakaljev et al. [14] presented specification data for the known small turbojet engines that had been produced in the past. The Teledyne J408 was an example of a single-shaft turbojet engine with a TIT of 1039 K and a nominal RPM of 42,000 r/min; the nominal thrust of this engine was 4.45 kN and the engine was equipped with a single-stage axial turbine.

Another model, the Teledyne 305-4, was a type of small single-shaft turbojet engine. However, this engine was much smaller than both the J408 and J402. Its maximal TIT was 1364 K and its maximal RPM was 81,000 r/min; due to its volume, it generated 0.18 kN of thrust. This engine was equipped with a single-stage axial turbine.

The last engine from the Teledyne company was the model 312. Its TIT was 1100 K and maximal RPM was 71,000 r/min. This engine was also equipped with a single-stage axial turbine. However, no cross-section images could be found. The maximal thrust of this engine was 0.79 kN.

Another interesting engine was the Williams International F107 series engine. Despite the fact that no cross-section images are available, it is known that it was a twin-shaft engine with high-pressure (HP) and low-pressure (LP) turbines. The HP turbine had a single stage with a maximal RPM of 64,000 r/min, while the LP turbine had two stages with a maximal RPM of 35,000 r/min; the maximal TIT was 1282 K and it produced 2.67 kN of thrust.

Another type of Williams International engine was the YJ400 series engine. It was also a type of bypass engine with both HP and LP turbines; however, both the HP and LP turbines had two stages. This engine was a twin-shaft engine, with a maximal RPM of 60,000 r/min, a TIT of 1227 K, and produced thrust measuring 0.89 kN.

Noel Penni was another manufacturer that produced well-known small turbojet engines. Their first model was the NPT 301. This engine had a single-stage axial turbine, with a maximal TIT of 1130 K and a maximal RPM of 40,500 r/min; it produced thrust measuring 1.37 kN. The model NPT 754 was similar to the NPT 301, but its maximal RPM was 34,100 r/min, with thrust measuring 5.09 kN.

The Microturbo company also produced two models of turbojet engines. The first was a single-shaft model TRS 18-046, with a maximal TIT of 923 K and RPM measuring 47,000 r/min. The second one was the single-shaft model TRI 60-1, with a maximal TIT of 1198 K and a RPM of 28,500 r/min; the thrust of these engines measured 0.9 kN and 3.43 kN, respectively.

## 2.2. Review of Small Turboshaft Engines

Small turboshaft engines are frequently used in aircraft to start the main engines and to produce electrical energy. Sometimes such engines are also used to provide auxiliary air for the aircraft; thus, they are equipped with a power output shaft. Nowadays there are many manufacturers of such engines around the world, most of which are located in the USA or Russia.

Radikovic and Jevgjenievic [15] reviewed every Russian-made small turboshaft unit up to the year 2020. In their work, the AI-450, RU-19A-300, TA-6A, TA-8V, TA-12-60, AI-8, AI-9W, and VSU-10 units were mentioned.

The AI-450 is a type of two-shaft auxiliary engine with a compressor turbine and a free turbine [16]. Its maximal TIT is over 1200 K, with a maximal RPM of 40,000 r/min and a 294 kW power rating ($N_{max}$). Additionally, the maximal mass airflow rate for this engine is 1.127 kg/s and the reduction in the gearbox's gear ratio is 6.5:1.

A cross-section of this engine is shown in Figure 1, where the reduction gearbox and the free turbine can be seen.

Another type of engine that might be suitable for the first stage of a TC unit is the RU-19A engine [17]. It is a single-shaft engine with a single-stage axial turbine and a 7-stage axial compressor. Little is known about its maximal TIT, however the maximal temperature after the turbine is 1173.15 K. Also, the maximal RPM is 16,000 r/min, with a $N_{max}$ of 720 kW. The turbine drives an AC generator via a set of bevel and helical gearboxes; the gearbox gear ratio is 0.447. The engine has its own fuel and lubrication system and the lubrication oil pressure varies from 1.2 bar at idle to 4.0 bar at 80% of the engine's maximal speed. The cross-section of this engine is shown in Figure 2.

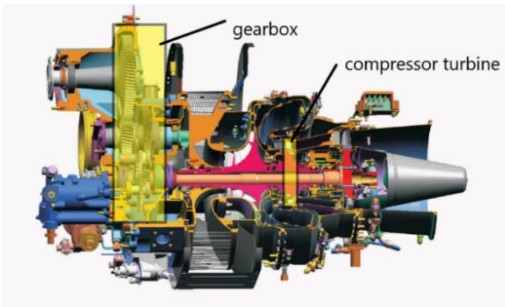

**Figure 1.** Cross-section of the AI-450 engine [16]. (Reproduced with permission from Ivchenko-Progress, AI-450M Turboshaft gas turbine engines; published by Ivchenko-Progress, 2012.)

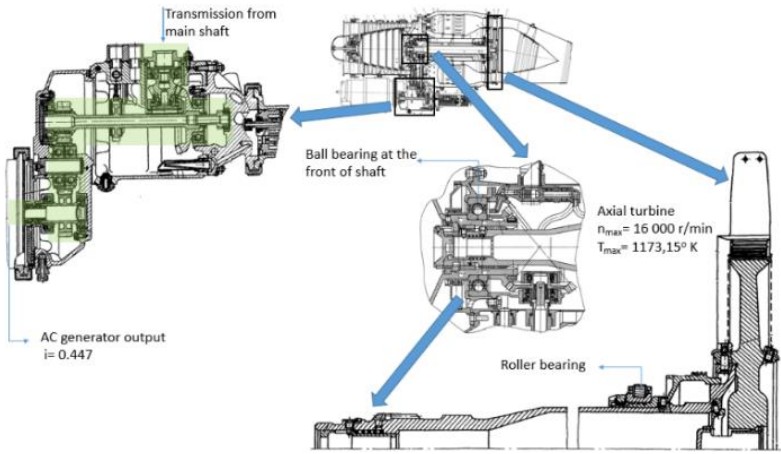

**Figure 2.** Cross-section of the RU-19A engine (Adapted from [17]).

The TA-6A is a single-shaft engine with a 3-stage axial turbine and a 3-stage compressor [18]. Its maximal TIT is 1110 K at a RPM value of 29,950 r/min, with a maximal $N_{max}$ of 45 kW; its maximal mass airflow rate is 1.35 kg/s. The turbine drives accessory devices and an AC generator via a helical gearbox. The reduction ratio of the gearbox to drive the generator is 0.2505 for the AC generator. The engine, with its multi-stage turbine, is very promising for TC applications. Its cross-section is shown in Figure 3.

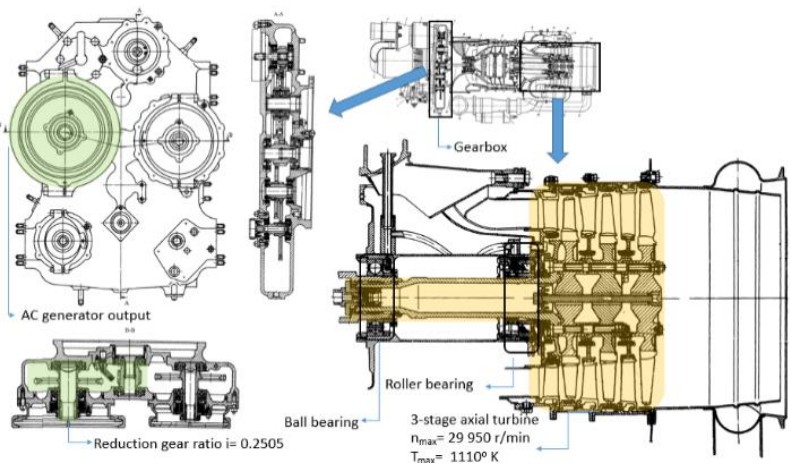

**Figure 3.** Cross-section of the TA-6A engine (Adapted from [18]).

A single-shaft TA-8V engine with a radial inflow turbine wheel was another option for the first stage of TC [19]; its maximal TIT and RPM are 1100 K and 40,139 r/min, respectively. The $N_{max}$ of this engine is 30 kW and the maximal mass airflow rate is 0.85 kg/s. The turbine drives an AC generator via a planetary gearbox. The gear ratio to drive the AC generator is 0.14952 and the engine has its own fuel and lubrication system. The required oil pressure is 4.5 bar, while the minimal pressure should not be lower than 1.2 bar. The cross-section of this engine is shown in Figure 4.

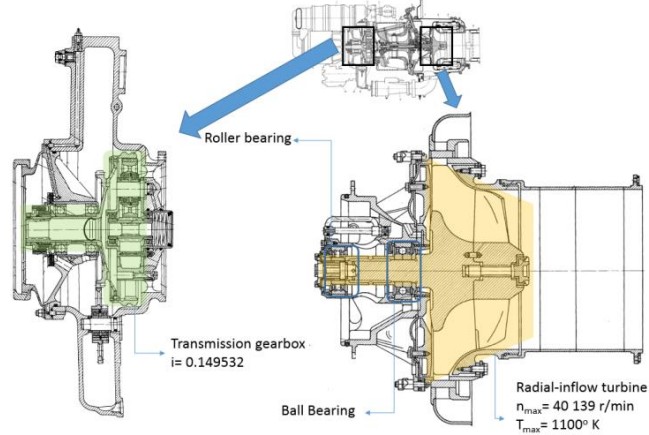

**Figure 4.** Cross-section of the TA-8V engine (Adapted from [19]).

Another option is the single-shaft TA12-60 unit, which has a three-stage axial turbine with a maximal RPM of 24,470 r/min and a TIT of 1115 K [20]. The maximal mass airflow is 2.1 kg/s, with a $N_{max}$ of 60 kW. However, it was not possible to obtain a cross-section image of this engine.

The AI-8 and AI-9 are well-known engines that are used in helicopters. The first one is a two-shaft engine with an axial compressor turbine and an axial free turbine. The TIT of this engine is 1200 K and its maximal RPM is 38,500 r/min. The reduction gearbox ratio to drive the AC generator is 0.25. Unfortunately, no cross-section images are available for this engine. The output power ($N_{max}$) of this engine is 14 kW.

As for the AI-9 engine, its single-shaft axial turbine is capable of withstanding 1100 K, its peak RPM is at 39,150 r/min, and its $N_{max}$ is 3 kW; its maximal mass airflow rate is 1.1 kg/s. The turbine drives an AC generator via a gearbox, with a gear ratio of 0.2. The cross-section of this engine is presented in Figure 5.

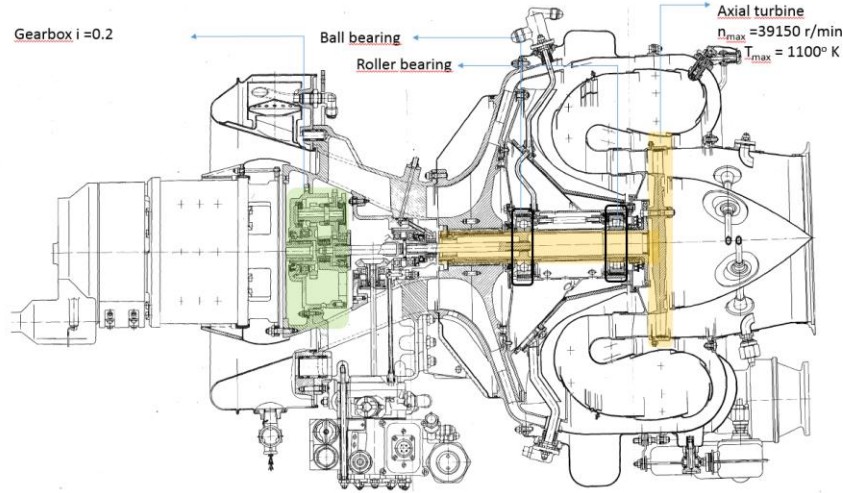

**Figure 5.** Cross-section of the AI-9W engine (Adapted from [20]).

The VSU-10 is a two-shaft engine with a 2-stage axial compressor turbine and a 2-stage axial free turbine [21]. The maximal TIT for the compressor turbine is 1050 K, with a maximal RPM of 31,975 r/min. The free turbine has a maximal RPM of 19,625 r/min and it drives an auxiliary compressor via the gearbox, with a reduction gear ratio of 0.62. The maximal mass airflow for this engine is 3.9 kg/s. The cross-section of this engine is shown in Figure 6. The compressor turbine has a roller bearing on either side, while the free turbine has a roller and ball bearing on one side.

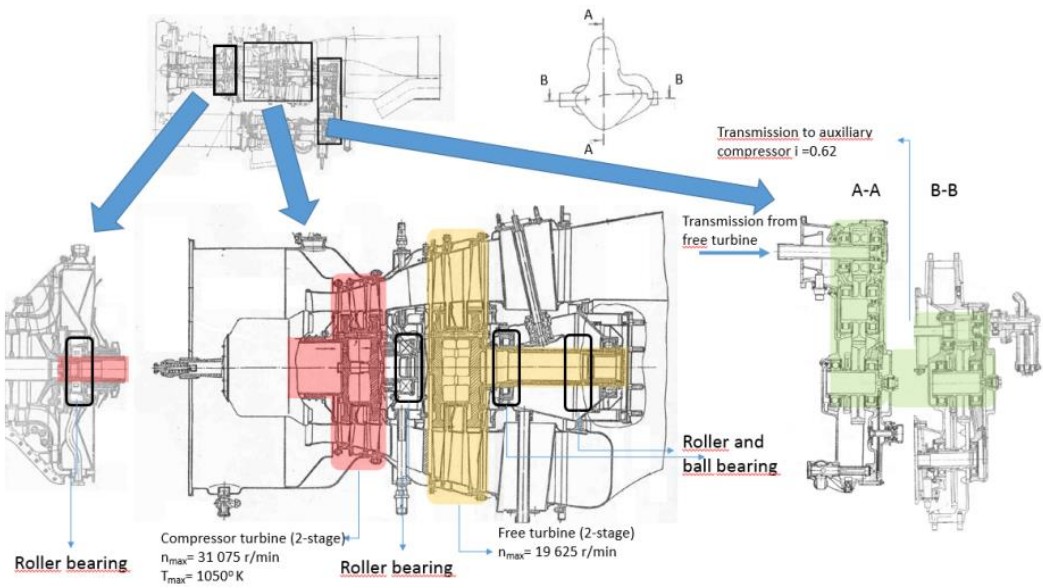

**Figure 6.** Cross-section of the VSU-10 engine (Adapted from [21]).

The last of the Russian small turboshaft engines is the GTDE-117 [22]. This unit is used to provide auxiliary power supply for the MIG-29 aircraft and to provide compressed air to start the main engines of the aircraft. Its cross-section is shown in Figure 7. It is a two-shaft engine with a compressor turbine and a free turbine; the maximal $N_{max}$ for this engine is 16 kW. The compressor turbine's maximal TIT is 1123 K, with a maximal RPM of 43,000 r/min. The maximal mass airflow rate for this engine is 1.0 kg/s. The compressor turbine has a roller bearing behind the turbine wheel and a ball bearing at the front of the turbine wheel, close to the place where the shaft couples with the compressor. The free turbine has roller and ball bearings on one side, and due to its short shaft, it is easier to balance.

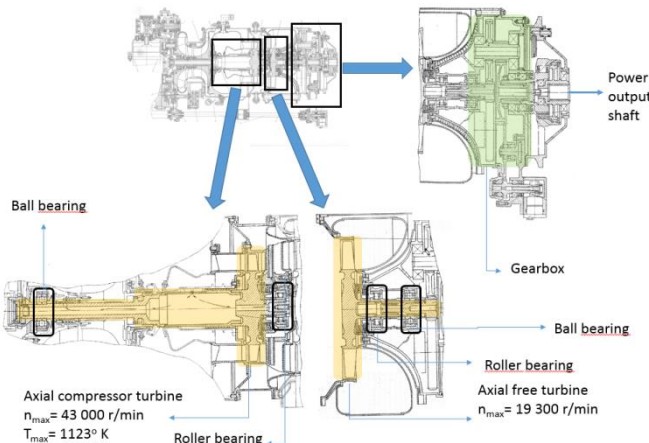

**Figure 7.** Cross-section of the GTDE-117 engine (Adapted from [22]).

Woodhouse [23] carried out a revision and comparison of the different models of small turboshaft engines from the United States manufacturers, such as Garrett, Honeywell, Pratt and Whitney, and others. A great amount of information was provided for the Garrett GTCP331 model, with its single-shaft and a 3-stage axial turbine. Different variants of the GTCP331 were produced, starting with the GTCP331-250, GTCP331-9 (A, B, D), and GTCP331-500. The last version produced the highest amount of power. The TIT for the GTCP331-500 was 1050 K, with a maximal RPM of 39,044 r/min and a $N_{max}$ of 169 kW. The highest value of the mass airflow rate was 3.53 kg/s. A cross-section of this engine is shown in Figure 8.

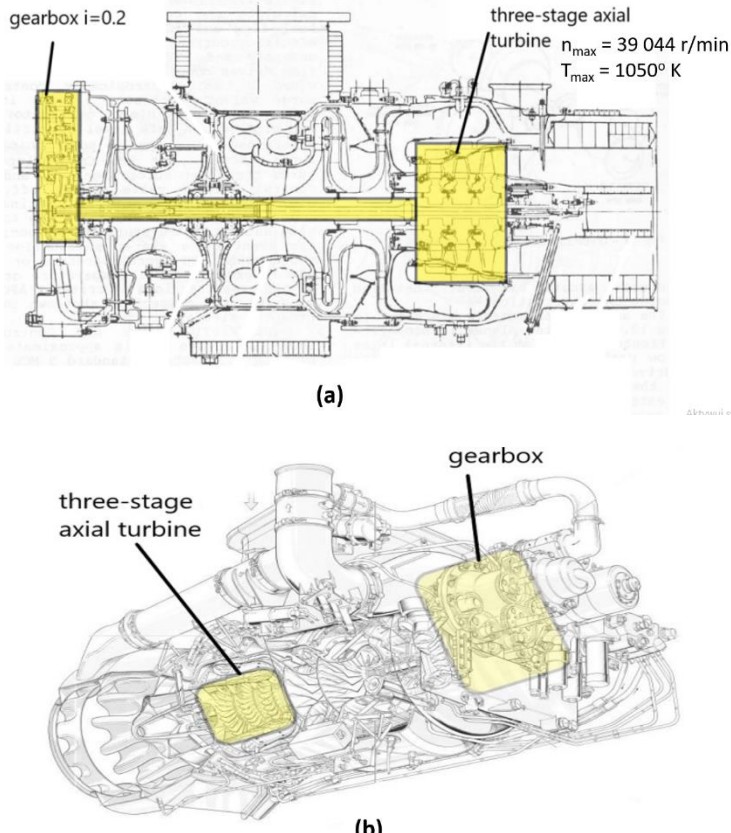

**Figure 8.** The GTCP331-500 engine: (**a**) cross-section; (**b**) isometric view (Adapted from [23]).

Another interesting engine is the Garrett AiResearch Model 85 [23]; however, very little is known about this engine. It is a single-shaft engine equipped with a radial inflow turbine and its maximal RPM is about 35,000 r/min.

Another engine is the Garrett Model 700 [23], which is a two-shaft engine with HP and LP axial turbines. An interesting fact is that variable guide vanes are installed before the LP turbine. However, no information about the performance of this model is available.

The next model is the Allied Signal RE220 [23]. This is a two-shaft unit with a 2-stage axial turbine. Its maximal TIT is 1055 K and its maximal RPM is 45,585 r/min. Its mass airflow rate is 1.22 kg/s and the $N_{max}$ is 66.2 kW. However, no cross-section images were available.

The model T-62T [23] is another engine that was widely used in military helicopters. It is a single-shaft engine with a maximal RPM of 61,565 r/min and a maximal TIT of 1000 K; the engine's $N_{max}$ is rated at 44.1 kW. This engine is equipped with a radial inflow turbine, as shown in Figure 9.

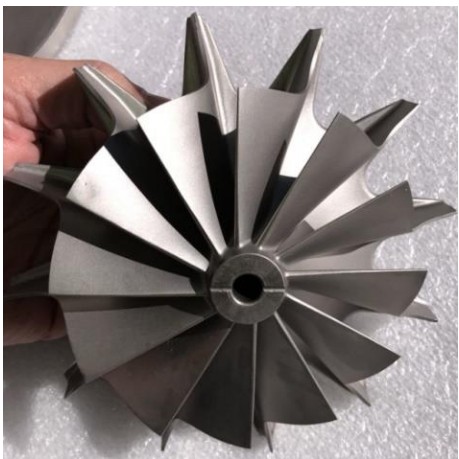

**Figure 9.** Turbine wheel from the T-62T engine.

Figure 10 presents an isometric view of an APS3200 engine's cross-section [23]. It is a two-shaft engine with a 2-stage axial turbine. The engine's maximal RPM is 49,300 r/min, with a maximal TIT of 1173 K. Additionally, the maximal mass airflow rate is 1.8 kg/s and the engine's $N_{max}$ is 91 kW. The turbine drives an AC generator via a gearbox, with a gear ratio of 0.4875.

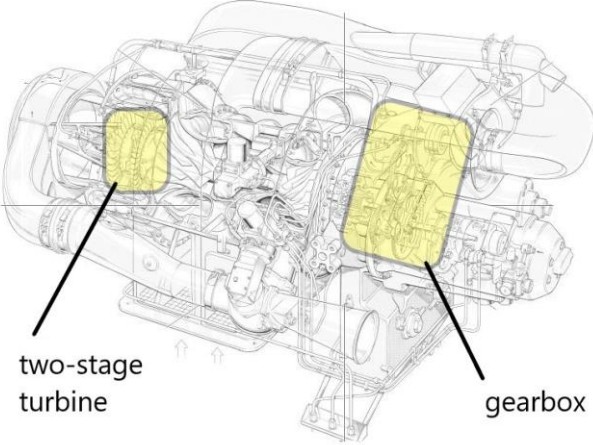

**Figure 10.** Cross-section of the APS3200 engine (Adapted from [23]).

There is a smaller version, which is the APS2300 [23]. This has a TIT of 1120 K, with a maximal RPM of 45,225 r/min, while the engine's $N_{max}$ is 44 kW. No cross-section images were available for this unit.

The APS5000 is the biggest of all the APS series [24]. Figure 11 presents this engine, which has a 2-stage axial turbine driving a centrifugal compressor and two DC generators. However, no data is available regarding its performance.

For this paper, a bigger GTD-350 turboshaft engine was also considered [25]. It is a two-shaft engine with a compressor turbine and a free turbine. For the compressor turbine, the maximal TIT is 1228.15 K and the maximal RPM is 45,000 r/min. It has a maximal mass airflow rate of 2.19 kg/s, which makes its compressor turbine wheel very promising for use as the first stage of TC; its maximal power ($N_{max}$) is 294 kW. The cross-section of this axial compressor turbine is shown in Figure 12.

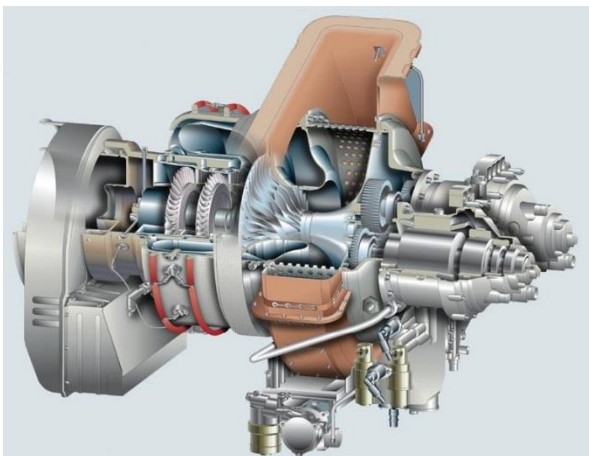

**Figure 11.** Cross-section of the APS5000 engine [24]. (Reproduced with permission from Pamintuan, E., MESA/Hamilton Sundstrand Training Academy; published by Mathematics Engineering Science Achievement (MESA), 2012)

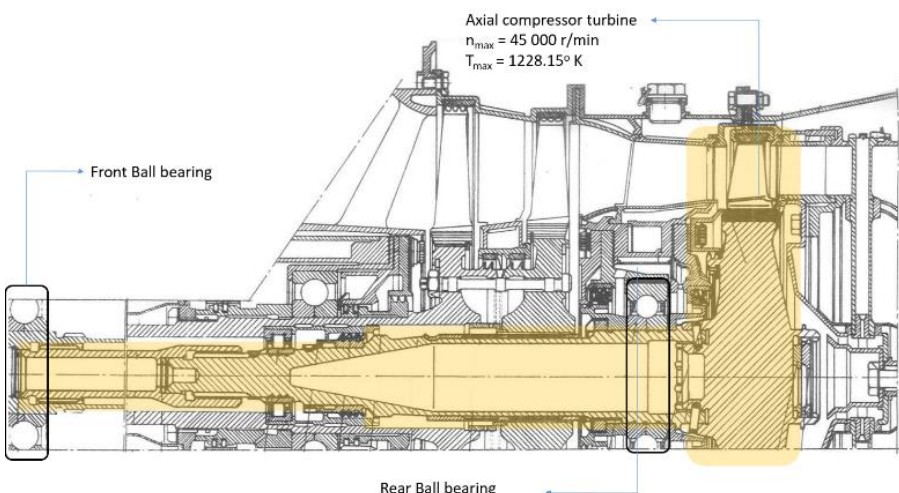

**Figure 12.** The GTD-350 engine, with a cross-section of the compressor turbine assembly (Adapted from [25]).

## 3. Results and Discussion

For the first stage of TC, small turbojet and turboshaft units were both considered. Figure 13 shows a comparison of the TIT and the $N_{max}$ of both small turbojet and turboshaft engines.

From Figure 13, it can be noted that the turbine wheels from turbojet engines such as the Noel Penni NPT754, the NPT301, the Teledyne J402, the 312, the 305-4, the Microturbo TRI60-1, the Williams YJ400, and the F107 are capable of withstanding temperatures above 1100 K, which is considered to be the temperature of the exhaust from a developed opposed-piston engine. Additionally, the maximal RPM of such engines is relatively high, which is desirable in exhaust turbines for automotive applications. The low TIT of the Microturbo TRS 18-046 engine (923 K) makes it less suitable for TC applications. Apart from the Williams F107 and YJ400, all the engines have a single-shaft configuration, which makes it easy to disassemble and assemble them. The main drawback of every turbojet engine is the lack of an output shaft and a reduction gearbox. In order to make TC operational, the power output shaft for a gearbox should be designed and manufactured to transfer the mechanical energy back to the crankshaft.

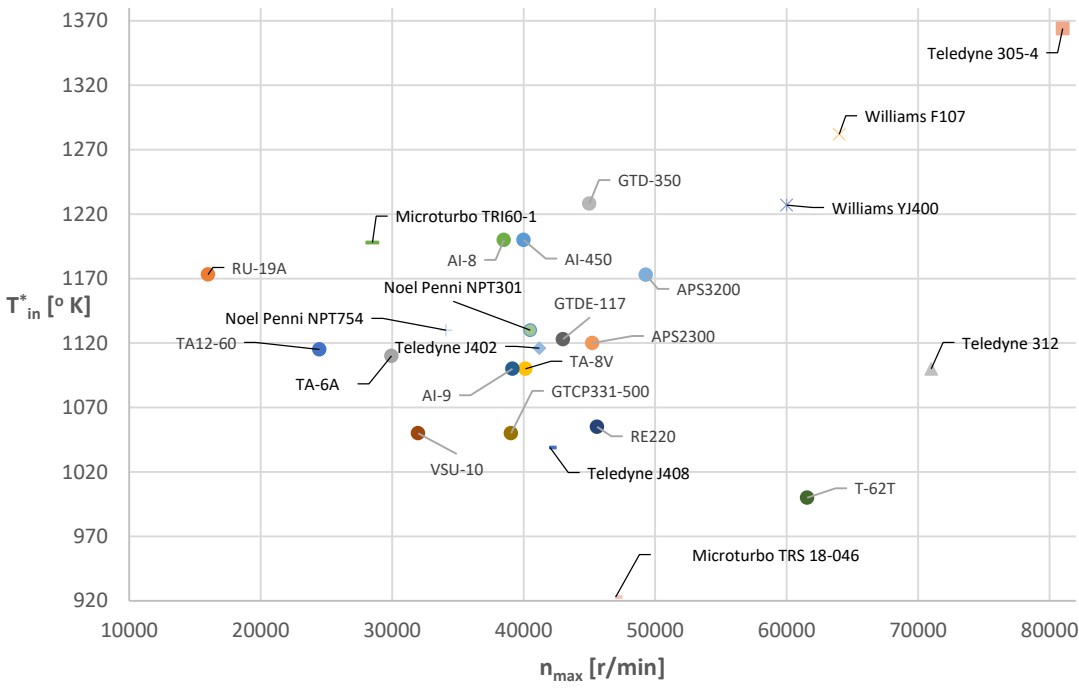

**Figure 13.** Comparison of the TIT ($T^*_{in}$) and $n_{max}$ of small turbojet and turboshaft engines.

Additionally, the small turboshaft engines TA12-60, TA-6A, TA-8V, AI-8, AI-9, AI-450, GTDE-117, APS2300, APS3200, and GTD-350 have a maximal TIT equal or greater than 1100 K and a high maximal RPM. Due to the lower TIT of the turbine wheels from the VSU-10, RE220, T-62T, and GTCP331-500 engines, they cannot be considered for the TC unit. Additionally, due to the low maximal RPM value of the RU-19A engine (16,000 r/min), such an engine cannot be considered for the TC unit. Small turboshaft engines are equipped with a power output shaft and a reduction gearbox, which makes it easier to adapt them for use in a TC unit. Figure 14 shows a comparison of the maximal TIT ($T_{in}$) and the maximal $\dot{m}_{air}$ of small turbojet and turboshaft engines.

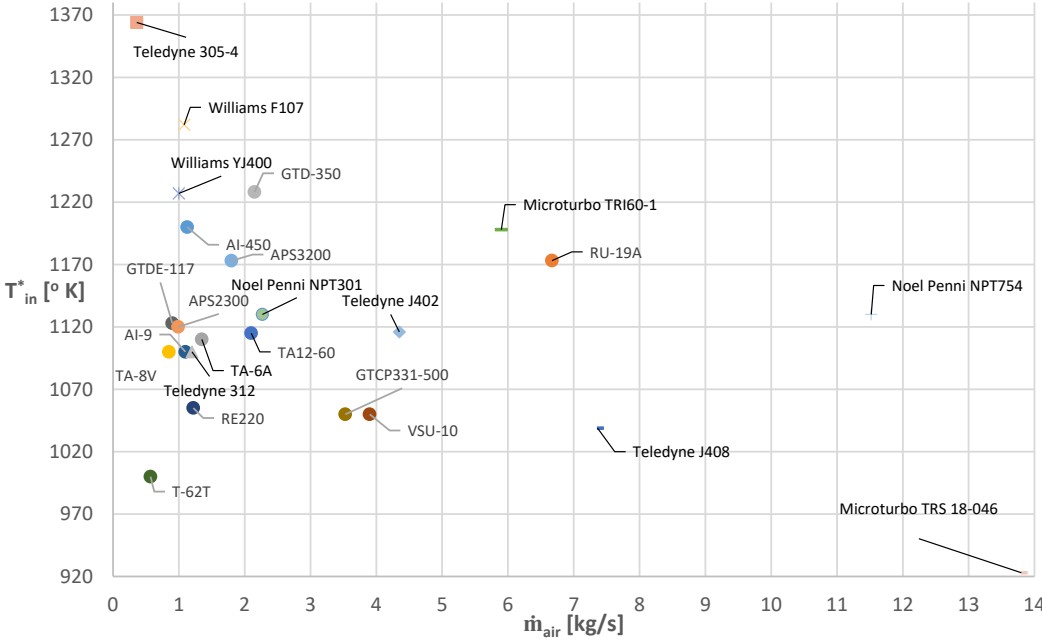

**Figure 14.** Comparison of the TIT ($T^*_{in}$) and $\dot{m}_{air}$ of small turbojet and turboshaft engines.

From Figure 14, it can be seen that the TA-8V, Teledyne 305-4, and T-62T engines operate at very low values of mass airflow rate. Such values are not acceptable for the TC unit, since the expected exhaust mass flow rate will be greater than 1 kg/s. With such a low mass flow rate, the exhaust gases would be choked during the exhaust phase.

## 4. Adaptation for OP Engine

The concept of an exhaust system with an adaptation of the turbine engines for the OP engine is presented in Figure 15. Such a system consists of a first-stage turbine, a turbine housing, a second-stage turbine, a hydraulic coupling, and a gearbox. The turbine housing has six inlets to prevent leakage of exhaust gases during the exhaust valve overlap. The turbine housing will be designed from the beginning and its configuration will depend on the first-stage turbine. The type of turbine impeller (radial or axial) for both first and second turbines depends on the chosen turbine engine. The first-stage turbine needs to withstand extremely high temperature from exhaust gases, which is why turbine engines with high TIT should be considered. Such a turbine should also withstand mass flow rates of at least 1.0 kg/s without choking. The second-stage turbine should be coupled with the gearbox and the hydraulic coupling in order to transfer torque back to the crankshaft.

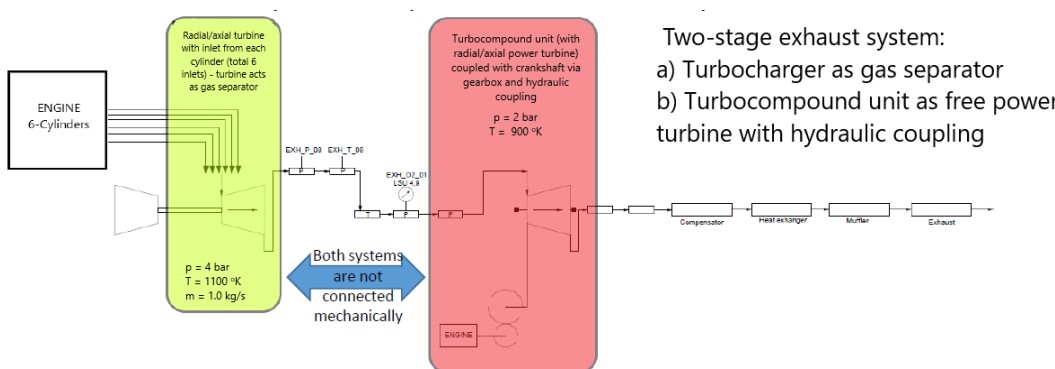

**Figure 15.** Conceptual adaptation of turbine wheels for an OP engine.

As can be seen from Figure 15, the first and second turbine stages will not be connected mechanically. Engines such as the APS500, APS 2300, APS3200, GTCP331-500, AI-9W, TA-6A, RU-19, J402, J408, TD 312, NPT 301, NPT 754, and TRI 60-1 units are single-shaft engines equipped with one HP turbine. Such configurations can only be addressed by the first-stage turbine. Additionally, for the second-stage turbine, an LP turbine from such engines as the GTD-350, GTDE-117, AI-450, AI-8, YJ400, and F107 units should be considered. Great care has to be taken considering the gearbox and the reduction gear ratio to ensure the turbine matches with the engine. Figure 16 shows the first-stage turbine from the GTD-350 engine that was obtained for the project. It is an axial-type turbine, the operating parameters of which are shown in Figure 12.

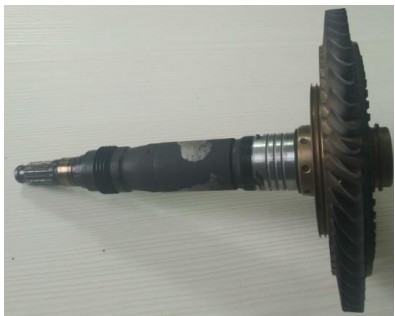

**Figure 16.** Compressor turbine from the GTD-350 engine.

This turbine wheel benefits from high mass flow rate and high efficiency; however, such efficiency is achievable within a narrow range of linear speed.

It was possible to obtain a K44 automotive turbocharger turbine. This turbine is shown in Figure 17, along with the scanning process and turbine model generated in the Unigraphics NX software.

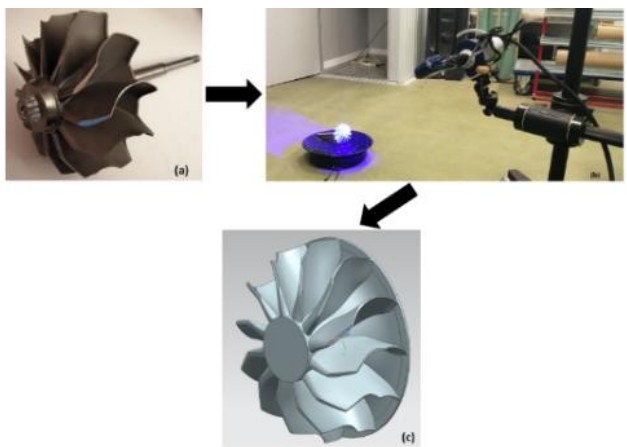

**Figure 17.** (**a**) Geometry of the K44 turbine, (**b**) turbine wheel scanned with the 3D scanner, and (**c**) the turbine wheel model.

## 5. Conclusions

To conclude, small turbojet and turboshaft engines were considered for automotive adaptation in an opposed-piston engine that is under development at Warsaw University of Technology. The adaptation aimed to recover waste heat energy from exhaust gases using a two-stage TC unit. The ORC system was considered in this paper.

- The literature review showed that parallel TC offered smaller BSFC reduction compared to series TC. This is attributed to the fact that in a parallel configuration, a greater amount of exhaust gases flow through a compressor turbine rather than a power turbine.
- Parallel TC produced smaller back-pressure compared to series TC, as the power turbine is not placed directly behind turbocharger. This position of the power turbine does not limit the pressure ratio of the turbocharger.
- In total, 25 turbine engines were compared, considering their maximal TIT, RPM, and $\dot{m}_{air}$.
- The Noel Penni NPT754, NPT301, Teledyne J402, 312, Microturbo TRI60-1, Williams YJ400, and F107 turbojet engines were considered appropriate for the TC unit. However, such engines are not equipped with a power output shaft, which has to be manufactured.
- The TA12-60, TA-6A, AI-8, AI-9, AI-450, APS2300, APS3200, and GTD-350 turboshaft engines were considered. These engines meet the requirements regarding the TIT, RPM, and $\dot{m}_{air}$. Additionally, these engines are equipped with a power output shaft and a gearbox. The output shaft and the gearbox can be used to transmit torque onto the crankshaft.
- The adaptation of the turbine engine into an ICE engine was presented, including the turbine wheels of the GTD-350 engine and the K44 radial inflow turbine.

## 6. Plans for Further Research

As mentioned in the previous sections, a two-stage TC unit will be used for the PAMAR V engine, which is a six-cylinder, opposed-piston engine with the cylinder axis positioned parallel to the main shaft. The engine is being developed at Warsaw University of Technology. The TC unit will be used as a waste heat recovery system, placed between the cylinders and the silencer. Additionally, the possibility of using automotive radial inflow turbine wheels for the two-stage TC unit was considered.

In the near future, an initial calculation of the first stage of the TC unit will be conducted, including numerical computational fluid dynamics (CFD) calculations.

**Author Contributions:** writing—original draft preparation, D.K.; writing—review and editing, P.M. All authors have read and agreed to the published version of the manuscript.

**Funding:** This research received no external funding.

**Acknowledgments:** The authors would like to acknowledge Military Aviation Works No. 4 in Warsaw for sharing the GTD-350 compressor turbine wheel. Additionally, the authors would like to acknowledge BorgWarner company Poland for sharing the K44 turbine wheels.

**Conflicts of Interest:** The authors declare no conflict of interest.

## Nomenclature

The following nomenclature is used in the manuscript:

**Symbols**

| | |
|---|---|
| $T_{in}$ | Turbine inlet temperature [$^{\circ}K$] |
| $\dot{m}_{air}$ | Air mass flow rate [kg/s] |
| $n_{max}$ | Maximal turbine revolutions per minute [$\frac{r}{min}$] |
| $N_{max}$ | Maximal engine power [$kW$] |

**Abbreviations**

| | |
|---|---|
| TC | Turbocompounding |
| ICE | Internal Combustion Engine |
| mech-TC | Mechanical Turbocompounding |
| el-TC | Electric Turbocompounding |
| HDD | Heavy-Duty Diesel |
| EGR | Exhaust Gas Recirculation |
| RCC | Rankine Combined Cycle |
| BSFC | Brake Specific Fuel Consumption |
| VTG | Variable Turbine Geometry |
| BMEP | Brake Mean Effective Pressure |
| ORC | Organic Rankine Cycle |
| TIT | Turbine Inlet Temperature |
| RPM | Revolutions Per Minute |
| HP | High-Pressure Turbine |
| LP | Low Single-Pressure Turbine |
| AC | Alternating Current |
| DC | Direct Current |

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
