# Peer review of "Review of Small Gas Turbine Engines and Their Adaptation for Automotive Waste Heat Recovery Systems"

_ijtpp, doi:10.3390/ijtpp5020008_

Round 1
Reviewer 1 Report
Can the authors comment in the introduction as to the suitability of turbocompounding versus the ORC system and why turbocompounding is still relevant?
Line 37-38: “At full-loads 1-stage turbocharging system provided 40.0÷74.8 kW of available energy of waste heat and 43.0÷76.3 kW for 2-stage turbocharging system”
What do you mean with these ratios? Do you mean 40.0-76.8kW?
A great number of English language mistakes exist which can be improved upon.
Section 2.1 what is the power rating (or thrust) of these engines. Include in text along with the information already provided from them
Line 114: why are you forced to look into gas turbines for modifications and not design a radial turbine from the beginning?
Overall, this is a practical guide to selecting small gas turbines for application in turbocompounding. It is more a technical report than a scientific paper.
Author Response
Dear Reviewer,
Thank You very much for Your comments and suggestions.
Turbocompounding was more suitable against ORC system for two major reasons:
Turbocompounding is less complex than the ORC system and thus it is cheaper in adaptation for the ICE engine. This was also a major factor for this project. The mass of turbocompound components is smaller compared to the mass of the ORC system.At lines 37-38, a single-stage turbocompounding system provided 40.0-74.8 kW of available energy of waste heat. The two-stage turbocompounding system provided 43.0-76.3 kW of available energy, depending on engine load and speed.
The power and thrust rating of the engines have been included in the manuscript.
Adaptation of turbine wheels that are already on the market is much cheaper than manufacturing new impeller. Some of the turbine engines are capable of withstanding required temperatures and are working within the required speed. This accelerates the design phase. However, designing a new wheel will be also considered in future research.

Reviewer 2 Report
My specific comments and questions are given below:
Major Comment: This publication is mainly a review of turbojet and turbo shaft engines and doesn’t discuss in detail their adaptation to diesel engine exhaust. A new section is required under results and discussion. This section should be dedicated how these turbines would be implemented to diesel engine exhaust. A feasibility study would be also interesting to support the discussion. Minor Comments: There are minor spelling and grammar mistakes, please correct them before publication. Include nomenclature and abbreviations sectionAuthor Response
Dear Reviewer,
Thank You for Your comments and suggestions.
Please find my reviewed manuscript with a section that is discussing the adaptation of the turbine engine into the Internal Combustion Engine.
With Regards,
Dariusz Kozak

Round 2
Reviewer 1 Report
A final check of English is required.
Author Response
Thank You very much for Your comments and suggestions.
Based on Your notes, I the paper has been reviewed by English native speaker. The paper's linguistic condition was improved. The improvement was focused on articles “a”, “an”, “the” as well as on minor spelling mistakes. The title of the paper was corrected and also, the references paper title format was corrected and the “sentence case” was used. I have also added DOI, ISBN and ISSN number to some of the references in my manuscript.
I have uploaded pdf file with indicated corrections.
With Regards,
Dariusz Kozak
